# Review of Intentional Electromagnetic Interference on UAV Sensor Modules and Experimental Study

**DOI:** 10.3390/s22062384

**Published:** 2022-03-20

**Authors:** Sung-Geon Kim, Euibum Lee, Ic-Pyo Hong, Jong-Gwan Yook

**Affiliations:** 1Department of Electrical and Electronic Engineering, Yonsei University, Seoul 03722, Korea; sgkim18@yonsei.ac.kr (S.-G.K.); ikanow@yonsei.ac.kr (E.L.); 2Department of Information and Communication Engineering, Kongju National University, Cheonan 31080, Korea; iphong@kongju.ac.kr

**Keywords:** UAV, drone, anti-drone, sensor module, intentional electromagnetic interference

## Abstract

With the advancement of technology, Unmanned Aerial Vehicles (UAVs), also known as drones, are being used in numerous applications. However, the illegal use of UAVs, such as in terrorism and spycams, has also increased, which has led to active research on anti-drone methods. Various anti-drone methods have been proposed over time; however, the most representative method is to apply intentional electromagnetic interference to drones, especially to their sensor modules. In this paper, we review various studies on the effect of intentional electromagnetic interference (IEMI) on the sensor modules. Various studies on IEMI sources are reviewed and classified on the basis of the power level, information needed, and frequency. To demonstrate the application of drone-sensor modules, major sensor modules used in drones are briefly introduced, and the setup and results of the IEMI experiment performed on them are described. Finally, we discuss the effectiveness and limitations of the proposed methods and present perspectives for further research necessary for the actual application of anti-drone technology.

## 1. Introduction

In the era of the 4th Industrial Revolution, drones have achieved innovative performance improvements based on new technologies, such as machine intelligence, networks, and sensors [1,2,3,4,5,6]. With the development of these technologies, drones can be operated reliably even in rapidly changing external environments; as a result, they are being used to perform various tasks that are difficult or avoided by humans, such as agriculture, disaster management, and fire detection [7,8,9,10,11,12].

However, with the increase in the use of drones, their illegal use, such as in terrorism and spycams, is also increasing [13]. As the number of crimes using drones has increased, there has been a growing body of research on anti-drone methods, including their detection, identification, and neutralization [14,15,16,17,18,19,20]. Among these methods, effective neutralization methods are essential to ensure reliable protection from actual threats. The methods of capturing or destroying drones by direct physical attacks, such as nets or gunfire are simple and straightforward; however, because of the fast speed of the target and possible secondary damages, they are avoided. Recently, neutralization methods using electromagnetic interference have been extensively studied to overcome these limitations. Most of the early studies were limited to the analysis of drone’s response to high-power electromagnetic waves [21]; however, recently, studies on electromagnetic waves with various frequencies and waveforms have been conducted [22,23,24,25].

For a stable flight, various sensors inside and outside the drone should operate in coordination. Therefore, if a few sensors are affected by the external interference, it can result in a serious malfunction. In addition, because sensor modules such as inertial measurement units (IMUs) are essential for most drones, introducing disturbances in these sensor modules is an appropriate method of neutralizing drones. Accordingly, recent studies have analyzed the response characteristics of the sensor modules to the interference signals in various experimental environments.

In this paper, we review the studies on electromagnetic interference in sensor modules, which can be used as an anti-drone method. In Section 2, we categorize the studies based on whether high-power signals are used, prior knowledge of the target is needed, and radio frequency signals are used. In Section 3, we investigate the sensor modules that are widely used in most drones and present a monitoring system that allows external access to the data used by sensor modules inside the drones. Section 4 describes the experimental studies conducted on the three categories of sensor modules presented in Section 2. Finally, Section 5, discusses the experimental results and the future research.

## 2. Review of Intentional Electromagnetic Interference

Electromagnetic interference that can affect electronic devices and systems varies widely from lightning to high-altitude electromagnetic pulses (HEMPs) caused by nuclear denotations. In addition, with the advancement of technology and easier access to various electronic equipment, the possibility of intentional electromagnetic interference (IEMI) has become common for malicious disturbances in electronic systems [26]. In this study, we focused on the IEMI that can affect commercial drones.

In the past, most studies on IEMI were limited to high-power electromagnetic interference (HPEM) using significantly high-power signals [27]. Therefore, the former categorization on electromagnetic interference had been only focused on high-power IEMI, while paying less attention to the low-power IEMI. The categorization was based on several criteria such as frequency coverage, underlying technologies, and effects on the target [28]. However, recently, much of the research has been focused on how to distribute limited power efficiently to achieve sufficient effect. Thus, researchers proposed the categorization method to encompass these works [29]. In [30,31], the author proposed the categorization of signal injection attack into regular-channel attack, transmission-channel attack, and side-channel attack according to the which channel of the target is being used. However, this categorization method could not convey the perspective of power as done in HPEM. So in this study, we classified electromagnetic interference into three categories: High-power IEMI, Low-power IEMI, and Non-RF. The target, method, and characteristics of each category are summarized in Table 1.
**High power IEMI:** High power IEMI uses an electromagnetic signal with sufficiently high power to destroy an electronic device or cause its serious degradation. Conventionally, the name high-power electromagnetic (HPEM) environment, which includes all naturally occurring electromagnetic phenomena, is mainly used. There is no exact criterion of power to be classified as HPEM; however, it is generally classified as HPEM if it has a field intensity above 100 V/m around the target [32]. In this study, to deal with the areas directly applicable to drones, all the methods of inducing destruction or degradation using high-power signals, without considering the operating mechanism of the target, are included by expanding the criteria.**Low power IEMI:** Low power IEMI uses an electromagnetic signal with relatively low power to cause malfunctioning of the target. The main difference from high-power IEMI is that low-power IEMI sets the applied signal in consideration of the target’s operating mechanism to lower the required power. Specifically, research has been conducted to couple an external interference signal into the sensor to manipulate the sensor’s data signal in a desired manner or to cause a malfunction. The targets of low-power IEMI can be classified as analog sensors, digital sensors, and communication modules. In general, analog sensors use low-voltage signals of a few millivolts. Thus, the data inside the sensor may be changed even when using a small number of coupled signals. To achieve this, an antenna or coil can be used as an attack method, and finding appropriate resonant frequencies to effectively couple the energy is important [33,34]. On the other hand, digital sensors use digital data with relatively high voltage levels of a few volts. Because a large amount of energy must be coupled to cause a significant change in the digital signal, a bulk current injection(BCI) method or direct power injection method (DPI) using an injection probe is commonly used rather than antennas [35,36]. For the communication modules using RF channels, if prior knowledge about the module’s channel information is available, in-band jamming may be an effective interference method, and it requires relatively low power.**Non-RF:** For electromagnetic interference, not only radio-frequency signals but also acoustic [37] and light signals [38] can be candidates for interfering signals. Because these methods operate by targeting the mechanical structure or operating mechanism of particular sensors, their versatility is low; however, they have a significant effect if the required conditions are met.

### 2.1. High Power Intentional Electromagnetic Interference (High Power IEMI)

High-power IEMI is a method of electromagnetic interference that uses high-power electromagnetic waves without specifically considering the operating mechanism of the target. This method has high versatility because it uses sufficient power to affect most of the electronic equipment, even without specific information of the target. However, because it uses very high power, detailed attacks, such as manipulating the target’s data in the desired manner are not possible. The targets, radiator types, and signal types used in high-power IEMIs are summarized in Table 2.

Pulse and continuous waves are possible candidates for high-power IEMI waveforms. From the perspective of the coupling mechanism, it is more effective to use a pulse signal with a wider spectrum than a continuous wave with a single frequency; however, it is highly complicated to build a system that generates pulsed signals. Refs. [39,40,41] used a pulse signal as the interference source; however, Refs. [42,43,44] demonstrated that continuous waves can also act as interference sources. In [39], a facility was designed for the high-power electromagnetic interference test, and the results of the experiment using the test facility were summarized. The test facility was designed for both outdoor and indoor tests. The system consisted of five microwave sources at fixed frequencies in the L, S, C, X, and Ku radar bands. It had a pulse compression system (PCS) to generate a pulse with a peak field strength of a few kilovolts per meter and pulse repetition frequency of 300 Hz to 1 kHz. The test results showed that the effects of interference were more prominent at a lower frequency and upset started around a few hundred volts per meter, while permanent damage occurred around 15–25 kV/m.

In [40], the authors tested the impact of an ultra-wideband (UWB) electromagnetic pulse(EMP) on a commercial aerial vehicle. They identified the parameters that played a major role in the malfunctioning of the drone by analyzing the effects of three UWB-EMP radiators with different pulse-repetition frequencies and amplitudes. The results showed that the amplitude of the EMP signal played a more dominant role than the repetition rate of the pulse.

In [41], the authors investigated the impact of narrowband high-power electromagnetic pulses characterized by frequencies between 100 MHz and 3.4 GHz on a commercial quadcopter. They established a system that can monitor the sensor data of drones in real time and applied an HPEM ramp signal so that the amplitude of the pulse increases linearly. The exact value of the pulse amplitude was not disclosed. The magnetometer and motor control signals were directly affected at a certain amplitude, and the accelerometer and gyroscope responded to the vibrations due to the change in motor speed. From these results, it was confirmed that if a sufficient field strength is applied, a specific sensor can be confused, and building a system capable of monitoring the sensor inside the drone is necessary.

In [42], the authors conducted a high-power IEMI test using high-power pulses and continuous wave signals on the Bluetooth module, “MULLE” operating at 2.4 GHz band with a mounted temperature sensor. The pulse with 100 ns duration and a 1 ms time period had no significant influence owing to its low average power. However, with the continuous wave having a peak electric field of 0.24 kV/m 0.36 kV/m, the temperature sensor was affected at 2.25 GHz with 7 °C increase and the data loss owing to the communication error was observed at 2.45 GHz.

In [43], the authors proposed methods to obtain privileged access to the aircraft’s main microcontroller and gather information. Using the proposed monitoring system, a high-power IEMI test was conducted while illuminating the target with continuous waves in the 100 MHz–2 GHz frequency range and amplitude modulation at a rate of 1 Hz–20 kHz. As a result, unrealistic variations in sensor data, such as in the battery temperature sensor and vertical speed, were reported for a field magnitude of 55 V/m and higher.

### 2.2. Low Power Intentional Electromagnetic Interference (Low Power IEMI)

Low-power IEMI is an electromagnetic interference method that uses low-power signals. In addition to the power-level perspective, what distinguishes it from high-power IEMI is that it uses the interference signals according to the operating mechanism of the target. Based on prior knowledge of the target, attackers increase the efficiency of the attack by identifying the effective manner of coupling and applying signals in the intended manner. However, this method has low versatility because it requires prior information about the target. Research on low-power IEMI has been conducted in various fields; the targets, used methods, and types of signals are summarized in Table 3.

In [45], the authors conducted an RF-immunity test of a drone under electromagnetic interference signals simulating a TV-broadcasting signal and a GPS in-band interference signal. The test was conducted at an EM-field strength of 10 V/m in an anechoic chamber. The test results showed that the communication between the drone logic boards, IMU board, and navigation board was not affected; only the GPS module was affected by the in-band interference signal. These results indicate that a field strength exceeding a specific value is required to influence the data through out-of-band interference.

In [46], the authors analyzed the effects of interference signals on analog sensors. The purpose of this study was to manipulate the sensor reading by injecting a voltage into sensors using external signals, which is possible because analog sensors use low-voltage data signals. The authors proposed two signals for low-power IEMI: baseband and modulated signals. The baseband attacks target systems with a low-pass filter using a baseband signal, which is at an in-band sensor-data frequency. These attacks require relatively high power because at these frequencies it is not possible to normally couple to the circuits. The modulated-waveform attacks the target system without filters using signals in the MHz and GHz ranges because short lines in the circuit are likely to respond to high-frequency signals. For an effective attack, an attacker should set an attack signal in consideration of the resonant frequency of the circuit for increased coupling and demodulation that will occur inside the circuit. Based on this analysis, the authors experimentally verified that the baseband signals can be used to attack external electrocardiogram (ECG) and cardiac implantable electrical devices(CIEDs), whereas the modulation signals affect the microphones, Bluetooth headsets, and MTS300CB sensor boards. This research has demonstrated that it is possible to manipulate the sensor data of analog sensors using low-power IEMI; however, its application to digital sensors is difficult owing to the high voltage levels of digital signals.

In [47,48], studies were conducted on low-power IEMI for digital data signals. Unlike analog signals, digital signals have relatively high voltage values; therefore, sufficient power should be coupled to manipulate the data. Owing to the limitation of low coupling of the antenna method, methods such as bulk current injection (BCI) and direct power injection (DPI) using near-field probes have been mainly studied. These methods are not suitable for application to drones because physical separation is essential for signal injection.

In [49,50,51,52], studies on jamming and spoofing on various targets, such as GPS, cameras, and remote controllers (RCs), were conducted. Jamming and spoofing are methods to directly interfere with RF channels, in which communication proceeds with in-band interference, therefore degrading the quality of communication or manipulating the target in the desired manner by injecting fake signals. In [49], the authors proposed a scenario for hijacking drones using GPS spoofing, and verified it experimentally. In [50], the author conducted experiments on the jamming and spoofing of GPS, video, and RC controllers. In [51], jamming methods were classified into five categories: barrage jamming, tone jamming, sweep jamming, successive-pulses jamming, and protocol-aware jamming, and the performance of each category was compared through experiments. From the perspective of performance, protocol-aware jamming demonstrated the best performance, and considering the convenience of the system configuration, sweep jamming was found to be effective. In [52], the authors proposed a prototype sweep-jamming circuit for remote controller signals and analyzed its performance according to the bandwidth.

### 2.3. Non-RF Interference

Most high-power and low-power IEMI, as summarized in Section 2.1 and Section 2.2, use radio-frequency signals as a source of interference. However, depending on the target characteristics, these signals are often ineffective. Typically, in the case of digital-IC sensors, the dimensions of the circuit are too small to be coupled with an RF signal, and the voltage of the signal is too large to be affected. Recent studies have suggested the use of non-RF signals, such as acoustic signals or lasers, as interference signals. Table 4 summarizes the recent studies on non-RF interference.

In [53], the authors proposed a method to neutralize drones by degrading the performance of microelectromechanical system (MEMS) gyroscopes using intentional acoustic signals. It is well-known that MEMS gyroscopes have resonant frequencies owing to their structural characteristics, and through experiments on 15 types of gyroscopes, it was verified that seven of them had resonant frequencies within audible frequency range. Subsequently, the authors conducted real attack experiments with two types of vulnerable gyroscopes and verified that the performance degradation of gyroscopes led to an unstable drone flight. Furthermore, in [37], the authors analyzed the effects of acoustic signals on the MEMS accelerometer and demonstrated that the accelerometer data can be manipulated using appropriate acoustic signals.

In [54], the authors proposed an attack method using directed acoustic energy (shock wave) generated by explosive C-4, and verified it experimentally. This method has high versatility; however, it is destructive owing to attacks using shock waves itself, not the resonance inside the drone.

In [30,55], interference methods using light sources, such as lasers and projectors, were proposed. In [55], the authors used a laser and projector to attack optical flow sensors. Because the optical flow sensor uses visible light as the input, a method of distorting the sensor input using a laser and projector was proposed. Experiments were conducted in various environments, such as tiles, carpets, concrete, and grass, to verify that the validity of the method, except for grass, where optical flow exhibits excellent performance. In [30], the authors proposed two different interference methods targeting light detection and ranging (LiDAR) sensors using saturating and spoofing by relaying. The former forces the sensor to saturate by using a strong input signal, and the latter confuses the input of the sensor by mimicking the mechanism of the LiDAR sensor, which measures the round-trip time of the fired laser pulse to determine the distances.

## 3. Drone Sensor Modules

### 3.1. Sensor Modules in Drone

From a structural point of view, drones are equipped with various components, such as the main processors, power systems, sensors, and communication modules, to operate. Because the open-air environment where drones are operated is dynamic and unpredictable, the main processor inside the flight-controller board periodically collects data from various sensors and controls the drone adaptively. Figure 1 shows a block diagram of the drone and the signal flow between its components.

Among the electronics inside a drone, the flight-controller board plays the most important role in its operation. It consists of various sensors that are essential for the operation of the drone, and other sensors can be added to improve its performance. In this study, the Pixhawk 4 board was used as the flight controller board. It consists of an inertial measurement unit (IMU) composed of a three-axis gyroscope, three-axis accelerometer, three-axis magnetometer, and barometer. In addition, a camera, GPS, and optical flow sensor can be added to improve the performance of the drone. The sensor modules mainly used in drones are as follows:**Inertial Measurement Unit (IMU):** IMU is an essential sensor module for navigation devices; it is a combination of a three-axis gyroscope and a three-axis accelerometer, and in some cases, a three-axis magnetometer is added to improve its performance. The acceleration measured by the accelerometer and the angular velocity measured by the gyroscope are combined to precisely estimate the attitude of the drone. The resulting data from this sensor fusion are transmitted to the main processor of the flight controller in the form of a digital signal. Because the IMU plays a critical role in estimating the attitude of the drone, stable flight is not achievable if these signals are affected by external interference. To interfere with IMU-data signals, a high-power interference signal is needed because the digital data signals have a relatively high amplitude, i.e., in the range of few volts. The effects of high-power electromagnetic interference on the IMU data signals are presented in Section 4.1.**Camera:** Cameras are the most widely used modules for drone use. A wide variety of camera modules are applicable to drones and can be divided into two types based on the communication scheme: analog and digital. Many custom drones are equipped with camera modules that use analog communication owing to their low price and high accessibility. Most commercial photography drones use a digital-communication camera module for stable communication with a large amount of data. The most effective way to interfere with the camera module through radio-frequency signals is to apply an in-band interference signal. The effects of the in-band interference signal on the camera module using analog and digital communication schemes are described in Section 4.2.**Optical flow:** The optical flow consists of a camera, various sensors, and a CPU. It films the video using an onboard downward camera and compares the subsequent frames to estimate the attitude and velocity of the drone. In this process, various sensors, such as sonar and gyroscope, are also used, and these data are processed by the on-board CPU. The resulting data are digital signals, which is the same as the IMU data. However, because the optical flow uses filmed data as the sensor input, an external light source may operate as an interference source by distorting the filmed image. Therefore, the effects of a laser attack on an optical flow sensor were examined and are described in Section 4.3**GPS:** The GPS module receives signals from satellites orbiting around the earth and calculates the actual position of the drone. To calculate the position accurately, more than four signals from separate satellites are necessary. Jamming and spoofing are well-known interference methods for GPS modules, and the relevant studies and experimental results are well known [56].

### 3.2. Sensor Data Monitoring

A sensor module, such as a camera, whose response characteristics can be easily grasped from the outside, is convenient for configuring an experimental setup for analyzing the response characteristics of the module to external interference signals. However, for sensor modules that exchange data internally, it is necessary to configure the system to monitor the sensor data externally. In this study, Pixhawk4, an open-source flight controller, was used to access these data. Using ‘QGroundControl(QGC)’ software, which is used as a ground control station(GCS) of Pixhawk4, we could plot the sensor data in real time, as shown in Figure 2. However, because the plotted graph changes in real time and cannot be saved directly, it is difficult to perform a detailed analysis. To overcome this problem, we saved the log files of the drone. Because these log files had ‘ulg’ file format, they were converted into ‘csv’ format using Python’s ‘pyulog’ library and subsequently analyzed. The sensor-data monitoring process is organized in Figure 3 as a block diagram.

## 4. Experimental Section

### 4.1. IMU Sensors Interfered by High Power IEMI Source

#### 4.1.1. Near-Field Measurement

As mentioned in Section 3.1, IMU is an onboard sensor in the flight controller board. The signals that the IMU exchanges with the main processor are digital signals of several volts, which is significantly higher than the signal voltages of the analog sensor, i.e., in the range of several millivolts. In addition, because the dimensions of the flight controller and the circuit inside are small, only a small amount of energy can be coupled into the circuit by an external interference signal. For efficient power coupling in the circuit, it is essential to select the resonant frequencies of the circuit. There are many ways to determine the resonant frequency of the board, such as measuring the coupled power by sweeping the frequency [46], finding the frequency which shows high radar cross section (RCS) [44], and near-field scanning to examine the electric field distribution [57]. Because of the small size of the flight-controller board and the limitations in setting up the experiment for other methods, we used near-field scanning to determine the resonant frequency of the flight-controller board. Near-field measurement result can be used for the estimation of the current in the circuit [58]. Therefore, by selecting a frequency having a large near-field magnitude, we could predict that the change in the internal current of the circuit due to the external interference signal will be large. The diagram and experimental setup of the near-field scanner are shown in Figure 4. The near-field scanner consists of a spectrum analyzer, probe positioner, and probe. The probe moves through predefined measuring points and saves the near-field data to generate a two-dimensional (2D) or three-dimensional (3D) electric-field distribution above the board. The scanning processes for the front and back sides of the flight controller were carried out from 30 MHz to 3 GHz while power was applied to the board and sensor.

Figure 5a shows the near-field scanning result of the flight-controller board. For both sides of the board, dominant electric fields are distributed between 30 MHz and 600 MHz. As shown in Figure 5b, the hot-spots of the electric field are the ribbon cable that connects the IMU to the back side and the main processor on the back side. What both parts have in common is that the data are actively transmitted and received, which could make them vulnerable to the HPEM attack. As a result, the frequency of the attack signal was set to 143.2 MHz and 256 MHz, which showed the highest electric field level.

#### 4.1.2. Interference Experiment

With the frequencies chosen in Section 4.1.1, interference experiments targeting the sensor data of the IMU were conducted. Figure 6a,b show the experimental setup schematic and realized experimental setup, respectively. The sensor data of the IMU in the flight controller were observed in real time using the sensor monitoring system proposed in Section 3.2. We used a BiLog antenna with a frequency range of 30 MHz–2 GHz and a typical gain of 6 dB in chosen frequency range. To increase the effect of IEMI, an amplifier with a gain of 50 dB was used together with the antenna for maximum possible output. Considering the loss of cables and connectors, the signal output used as the antenna input was approximately 50 dBm. The field intensity around the target was determined according to the antenna characteristics and the path loss between the antenna and target.

Figure 7 shows the gyroscope data when the IEMI signals with two different frequencies are applied. In both gyroscope and accelerometer, no changes in sensor data are observed at both the frequencies owing to the insufficient amount of coupling onto the circuit. The reason for this low coupling level is the small dimensions of the target circuit. To overcome this problem, considerably high power signals with broadband characteristics are required [39].

### 4.2. Camera Module Interfered by Low Power IEMI Source

The camera module of a drone can be classified as either analog or digital. For the comparison of the two categories, we chose a custom drone as well as a commercial drone (DJI Mavic Air2). The former uses a commercial camera module using an analog communication method, whereas the latter uses a camera module using Ocusync2.0, the manufacturer’s own communication protocol based on the OFDM scheme. In Section 4.2.1 and Section 4.2.2, as an example of low-power IEMI, in-band interference experiments performed using continuous wave and digitally modulated signals for camera modules using analog and digital communications are described. The changes in video quality due to external interference signals can be classified into three categories, and the results of the following experiments were evaluated based on the criteria summarized in Table 5.

#### 4.2.1. Custom Drone

The camera module used in the custom drone uses a 5.8 GHz ISM band with frequency modulation when transmitting the captured video to the receiver. During the experiment, we set the center frequency of the channel to 5.74 GHz. For jamming signals, we used a signal generator and standard gain-horn antenna. Figure 8a,b show the diagram and the experimental setup.

Table 6 shows the scenarios that we considered to analyze the effect of frequency and level of jamming signal on the quality of the received video. For Scenario 1, we swept the frequency while fixing the signal strength to 23 dBm, which is the maximum output of the signal generator. For Scenario 2, we swept the level of the interference signal while fixing the frequency to 5.74 GHz, which is the center frequency of the video-transmission channel. Finally, through Scenario 3, the influence of frequency was analyzed when the intermediate intensity, identified through the previous scenarios, was used.

Table 7 summarizes the results for the proposed scenarios. When a relatively high output was used, as in Scenario 1, degradation of video quality began when the difference between the center frequency of the channel and the attack frequency was 17 MHz, and the video was disabled when the difference was smaller than 14 MHz. When the attack frequency was set to the center frequency of the channel, as in Scenario 2, the degradation occurred when the attack signal was only −5 dBm, and the video was disabled when the signal of 10 dBm output power was applied. When an appropriate output signal was used, as in Scenario 3, by combining the previous two scenarios, the degradation started at a frequency difference of 8 MHz, and the video was disabled when the frequency difference was 5 MHz. In summary, a high-power output allows significant attacks even if the target’s channel frequency is not exactly known; moreover, if the target’s information is known, an effective attack is possible even with a relatively low output. Figure 9 shows the examples of received videos in each scenario.

#### 4.2.2. Commercial Drone (DJI Mavic Air2)

The camera module used in DJI Mavic Air2 uses a proprietary communication protocol called Ocusync2.0. It is a digital communication protocol based on OFDM, which occupies a wider channel bandwidth than the analog communication module described in Section 4.2.1. As a result, the continuous-wave signal described in Section 4.2.1 would not be a candidate for an effective interference signal. Therefore, for the barrage jamming [51], for blocking the sufficient portion of the channel, we generated an OFDM signal with a bandwidth of 20 MHz using a vector signal generator and amplifier. Figure 10a,b show a diagram of the experimental setup. We analyzed the quality degradation of the received video while applying the generated interference signal to the flying drone. Figure 10c,d show the realized experimental setup.

Table 8 summarizes the experimental scenarios. Before the experiment, the video-transmission channel of the drone was manually set to 2.42–2.44 GHz for ease of analysis. For Scenario 1 and 2, the level and frequency of the interference signal were swept. For Scenarios 3 and 4, the position of the receiver was altered to analyze the effect of the signal-path overlap.

It was observed that while increasing the interference signal level from 0 dBm, lagging of the received video was observed when the signal level reached 6 dBm. However, unlike the results of the previous analog-communication module case, distortion of the video was not observed. In Scenario 2, the center frequency of the interference signal was swept from 2.33 GHz to 2.53 GHz, which causes lagging as well as serious distortion of the received video signals. The transceiver of the camera module determines the transmission rate according to the SNR of the channel [59]. In Scenario 1, the SNR of the channel gradually decreased, so that the transceiver could adapt the transmission rate in advance, leading to the lagging video. However, in Scenario 2, the bit rate could not be adaptively reduced because the SNR of the channel was radically reduced. For Scenarios 3 and 4, the video-transmission path and interference signal path did not overlap, and the minimum influence was observed. The results of Scenario 1 and Scenario 2 are shown in Figure 11a,b.

As summarized in Table 9, signals with wideband modulated waveforms are quite effective in interfering with the video transmission of the OFDM-based camera module. The power of the interference signal should be sufficiently high to lower the SNR of the transmission channel, and it is effective in causing radical SNR variation to prevent the transceiver from adaptive operation.

### 4.3. Optical Flow Sensor Interfered by Non-RF Source

An optical flow sensor estimates the attitude and velocity information of the drone by periodically comparing the subsequent frames of the video filmed with a built-in camera. Accordingly, by temporarily distorting the image captured by the optical flow sensor, the drone may misrecognize the attitude and velocity information, which can lead to a possible malfunction. To verify this scenario, we conducted a simple experiment using a commercial laser pointer. The experimental setup is illustrated in Figure 12a,b. With the optical flow sensor connected to the laptop through the flight controller board, the laser was irradiated on the camera lens of the sensor. The laser pointer used for the experiment had a wavelength of 650 nm and output power of 1 mW. Figure 12c shows the image filmed with an optical flow sensor without a laser attack, and Figure 12d shows the image when the optical flow sensor is attacked with the laser. It is clear that the filmed image is severely distorted by the laser, and it can be inferred that the information delivered by the optical flow to the flight controller board is also significantly distorted.

Through the monitoring system described in Section 3.2, the changes in pixel flow and gyro data transmitted by the optical flow sensor to the flight controller board were confirmed. Figure 13a shows the changes in the gyro data in the x, y, and z directions when the laser attack is applied twice, and the corresponding gyro compensation data, which is the opposite of the gyro data, is used to compensate for the undesired change in the gyro data. The flight-controller board may receive this compensation data and force the other control unit to compensate for ’fake instability’ caused by laser attacks, which can lead to malfunctioning of the drone. Figure 13b shows the changes in pixel-flow data in the x and y directions and the corresponding pixel-flow compensation data, similar to the gyro data.

The results showed that by irradiating the optical flow sensor with a laser, it is possible to distort the filmed image, which causes meaningful changes in the optical-flow-sensor data. If the drone was flying with an onboard optical flow sensor, the generated compensation data would have caused serious malfunctions, such as rotation, acceleration, and crash. However, because it is very difficult to accurately focus the laser on the sensor armed on drones during flight, research on the types of light sources and scenarios applicable to drones in flight is necessary.

## 5. Conclusions

In this paper, we reviewed various studies on the effect of intentional electromagnetic interference (IEMI) on sensor modules, which may be used for anti-drone purposes. Most of the previous anti-drone methods were limited to jamming and spoofing for GPS or RC signals; however, in this study, the scope was expanded to the intentional electromagnetic interference on sensor modules used in most drones. In this study, three categories of IEMI were identified: high-power IEMI, low-power IEMI, and non-RF, based on the presented criteria. Additionally, experimental studies were conducted for each category with experimental setups that were realizable in the laboratory.

The modules selected for the experiments were the inertial measurement unit (IMU), camera, and optical flow sensor. For the IMU, a high-power continuous-wave signal with frequencies chosen by near-field measurements was applied, but no significant change in the sensor data was observed. In addition, better effects may be obtained using the interference signal of a high-power pulse having a broadband spectrum. An in-band jamming experiment was conducted using the camera. The camera module using the analog communication scheme was significantly affected by the continuous wave signal, whereas the camera module using the digital communication scheme was affected by the digital modulated signal. In both the cases, the in-band jamming was effective when the signal to noise ratio (SNR) of the video-transmission channel degraded. In particular, for the camera module with digital communication protocol, a rapid SNR change would be effective because the transceiver could not adapt to the rapid change. For the optical flow, an experiment using a commercial laser pointer was conducted. With an external laser attack, significant changes in the sensor data were observed, and the effectiveness of the attack was confirmed from the corresponding compensation data. Further extensive experimental studies under various environments based on the studies and experiments reviewed in this paper are needed for practical application of anti-drone systems.

## Figures and Tables

**Figure 1 sensors-22-02384-f001:**
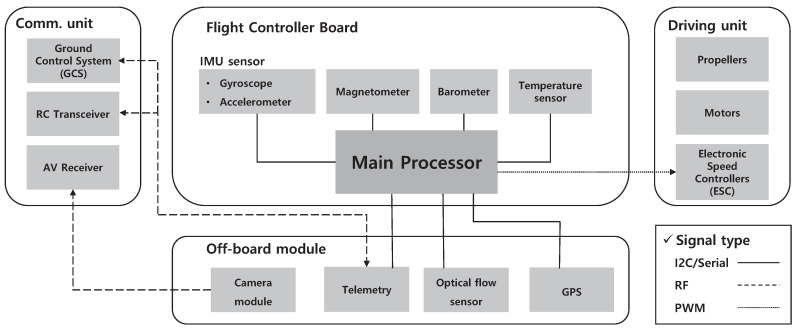
Block diagram of various modules in drone and the signal flow.

**Figure 2 sensors-22-02384-f002:**
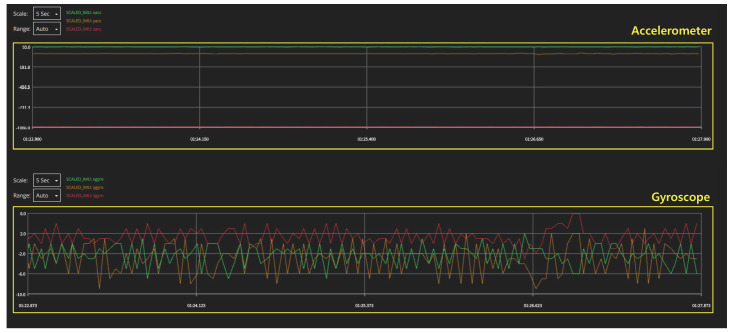
Real time sensor data monitoring using ‘QGroundControl’ software.

**Figure 3 sensors-22-02384-f003:**
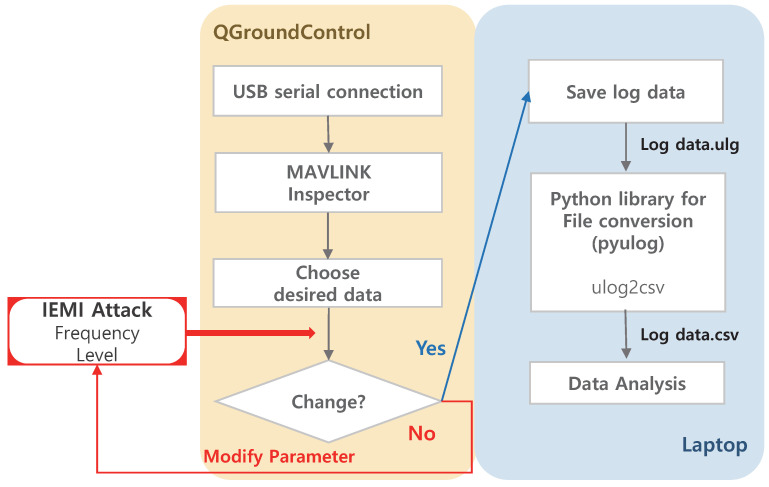
Block diagram for sensor data monitoring using log files.

**Figure 4 sensors-22-02384-f004:**
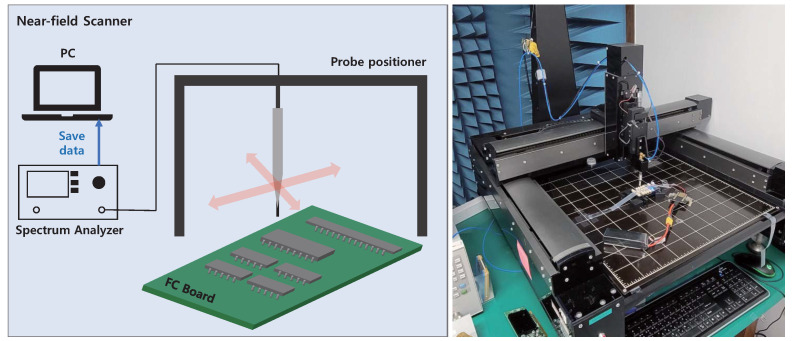
Diagram and experimental setup of near-field scanner.

**Figure 5 sensors-22-02384-f005:**
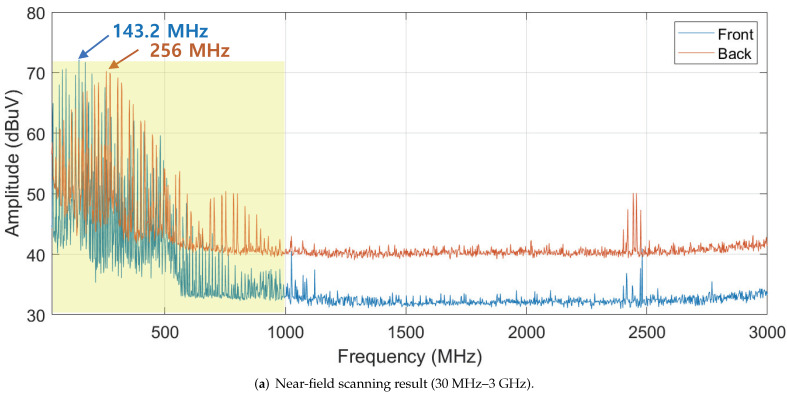
Result of flight controller near-field scanning. Both the front and back sides were scanned, and the probe tip is effective from 30 MHz to 3 GHz. (**a**) Near-field scanning results showing peak value of each frequency. (**b**) 2D electric-field distribution at the frequency with the highest peak value for each aspect. (Front: 143.2 MHz, Back: 256 MHz).

**Figure 6 sensors-22-02384-f006:**
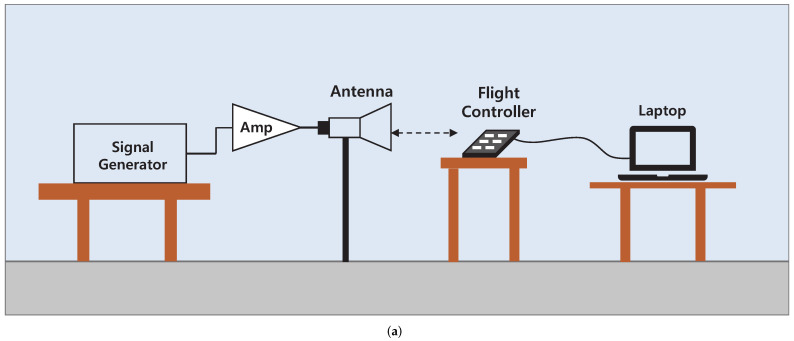
Experimental setup for high power IEMI experiments. (**a**) Experimental setup diagram; (**b**) Realized experimental setup.

**Figure 7 sensors-22-02384-f007:**
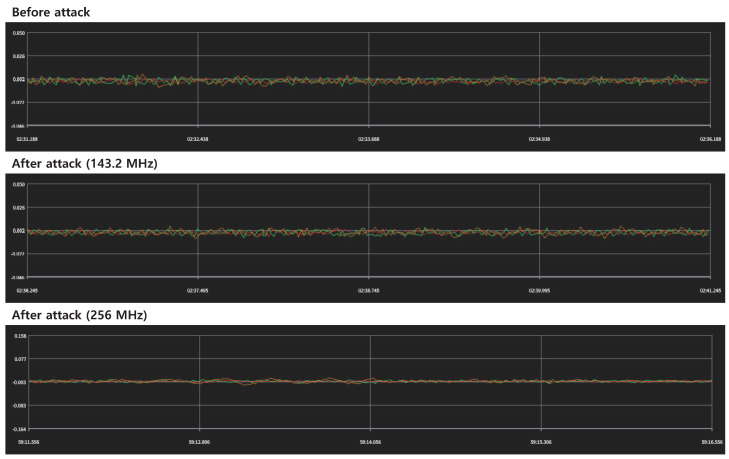
Monitored gyroscope data with high power IEMI attack.

**Figure 8 sensors-22-02384-f008:**
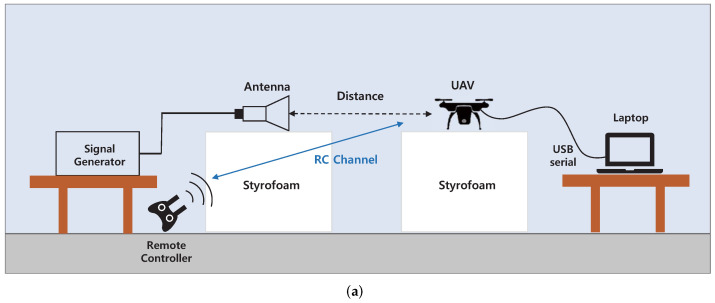
Experimental setup for low-power IEMI experiment on custom drone (Pixhawk 4). (**a**) Experimental setup diagram; (**b**) Realized experimental setup.

**Figure 9 sensors-22-02384-f009:**
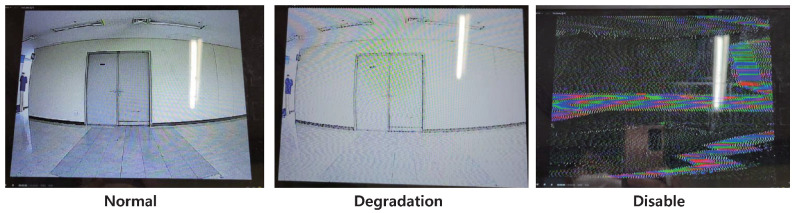
Examples of received videos in each status.

**Figure 10 sensors-22-02384-f010:**
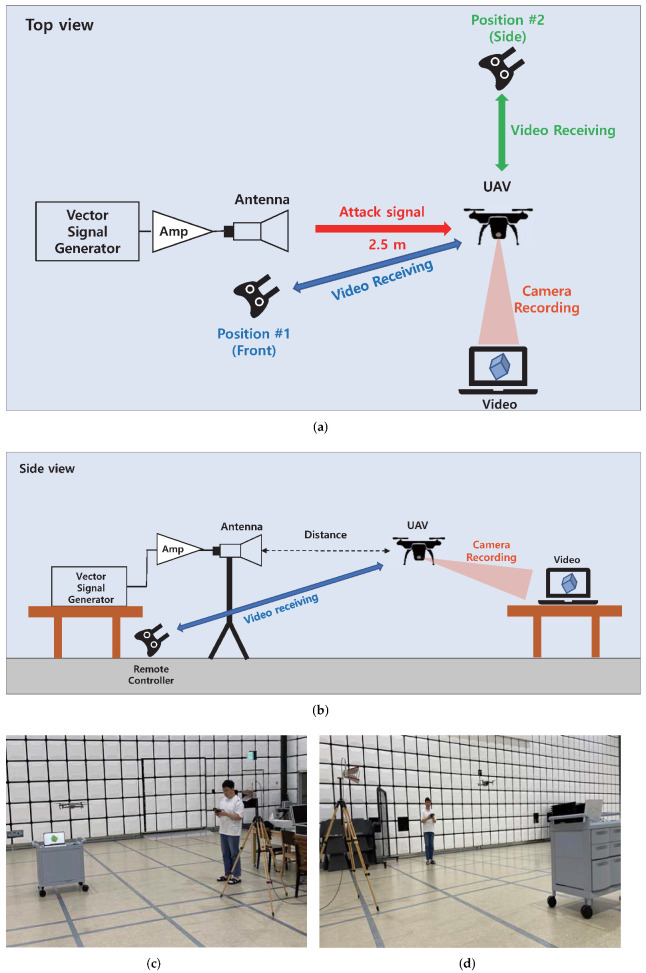
Experimental setup for low power IEMI experiment on commercial drone (DJI Mavic Air2). (**a**) Experimental setup diagram (Top view); (**b**) Experimental setup diagram (Side view); (**c**) Realized experimental setup (Front); (**d**) Realized experimental setup (Side).

**Figure 11 sensors-22-02384-f011:**
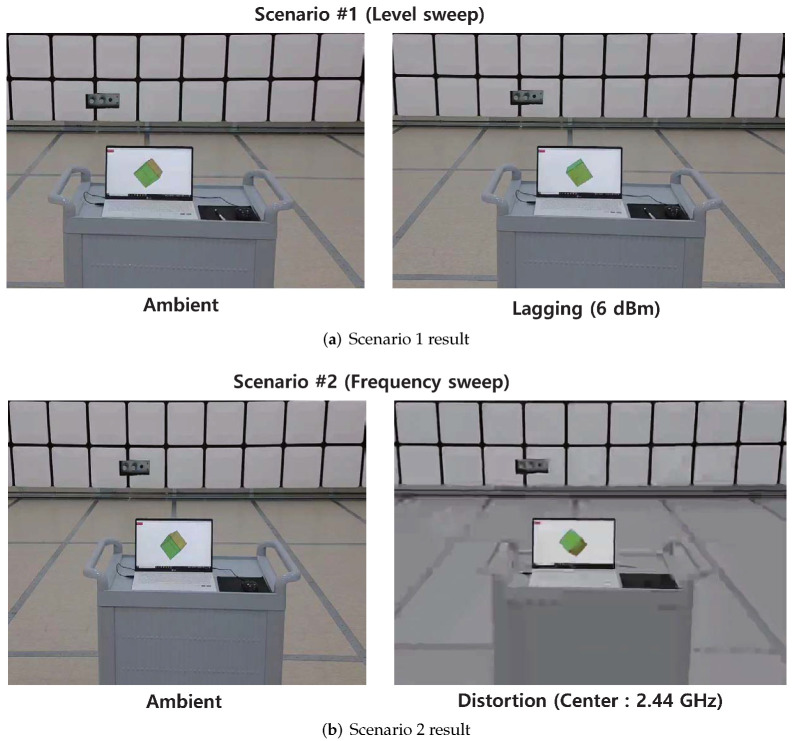
Results of low-power IEMI on camera module using digital communication scheme. The video status changed for Scenario 1 and Scenario 2. (**a**) Lagging occurred when the attack signal level exceeded 6 dBm. Distortions were not observed. (**b**) Distortion occurred when the attack signal overlapped with the video transmission channel. When the attack signal exceeded the channel frequency, the video quality was recovered.

**Figure 12 sensors-22-02384-f012:**
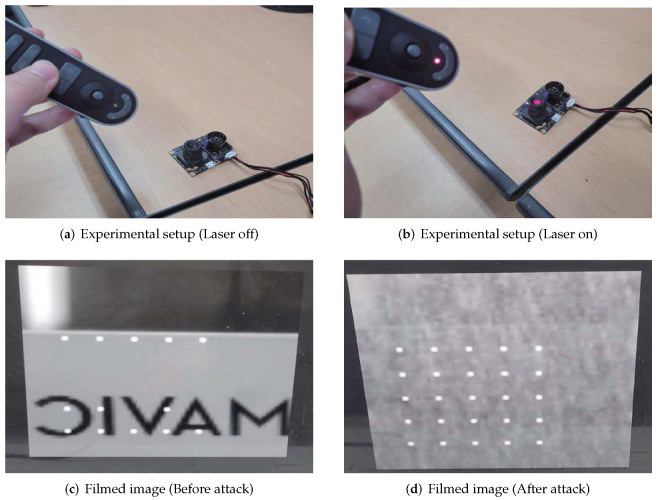
Experimental setup for non-RF IEMI on optical flow. (**a**) Laser attack off. (**b**) Laser attack on. (**c**) Filmed image before the laser attack was applied. To easily indicate changes, an object with written letters was placed in front of the camera. (**d**) Filmed image when the laser attack was applied. A distorted image was received.

**Figure 13 sensors-22-02384-f013:**
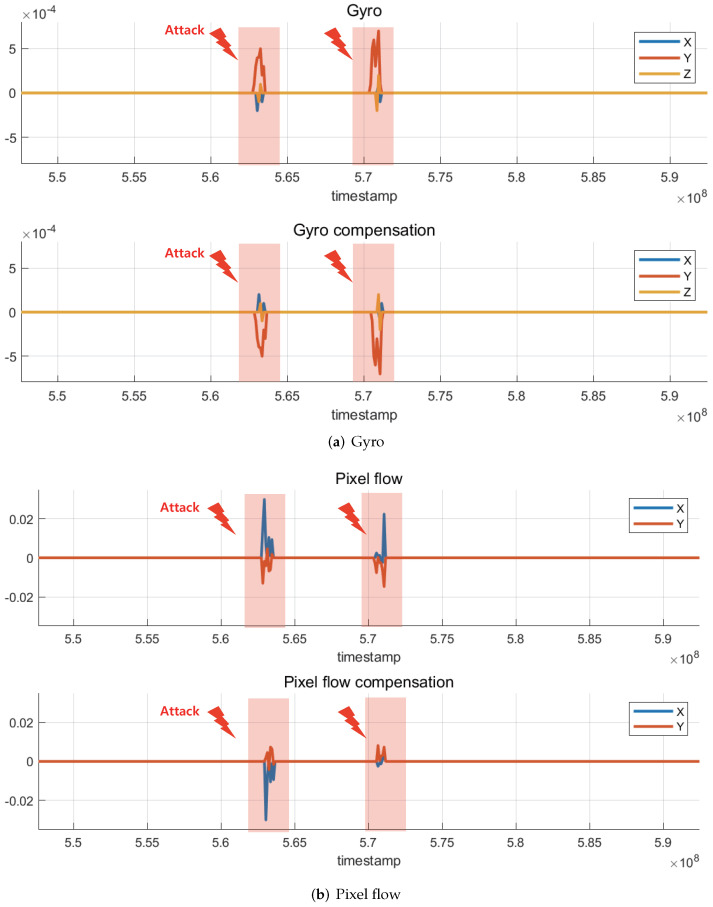
Monitored optical-flow sensor data. Abnormally calculated value led to the corresponding compensation value that might be critical for a stable flight. (**a**) When the attack was applied, the sensor calculated abnormal gyro data and the corresponding compensation value. (**b**) When the attack was applied, the sensor calculated abnormal pixel flow and the corresponding compensation value.

**Table 1 sensors-22-02384-t001:** Categorization of intentional electromagnetic interference.

Category	Main Target	Method	Specification
High Power IEMI	Electronic circuits	Antenna	- Destruction/degradation of the device using high-power EM wave
Low Power IEMI	Analog sensor	Antenna Coil	- Resonant frequency is needed for efficient coupling
	Digital sensor	BCI DPI	- Injecting the signal to the target at near distance with probes - Antenna is not applicable because of not enough coupling
	Comm. module	Antenna	- Prior knowledge of RF communication channel is needed - In-band jamming
Non RF	MEMS sensor	Acoustic	- Mechanical resonance of MEMS-based sensor
	Optical flow	Laser	- Degrading the received image of optical flow sensor - Degraded image leads to the malfunction

**Table 2 sensors-22-02384-t002:** Research on high power IEMI.

Ref.	Target	Radiator	Signal Type/Strength
[39]	Unprotected Electronic System	Cassegrain Antenna Gain: 37–40 dB	Pulse Peak field: few kV/m PRF: 300 Hz–1 kHz
[40]	Commercial drone (DJI Phantom 3)	UWB EMP Radiator (4 TEM-horns)	UWB EMP FOM = E (V/m) * R (m) (1.6 kV, 3.5 kV, 49.5 kV)
[41]	Commercial quadcopter	Horn antenna	Narrowband pulse 100 MHz–3.4 GHz PRF: 1 kHz
[42]	Sensor network (MULLE)	Horn antenna	Continuous wave 2–3 GHz Field peak: 0.24–0.36 kV/m
[43]	COTS Quadcopter	-	Continuous wave 100 MHz–2 GHz Field: 75 V/m–95 V/m

**Table 3 sensors-22-02384-t003:** Research on low power IEMI.

Ref.	Target	Method	Signal Type/Strength
[45]	Drone (Mikrokopter)	Radiated EMI	Continuous wave 470–862 MHz 1.4–2.7 GHz 10 V/m
[46]	Analog sensors (CIEDs, Mic)	Radiated EMI	- Baseband 0.1 Hz–1 kHz High power - Modulated Resonant frequencies Low power
[47]	Evaluation board (SASEBO-G)	Bulk Current Injection (BCI)	Injection probe 1–400 MHz 40 dBm
[48]	CMOS IC	Direct Power Injection (DPI)	Micro-probe 100–2000 μm −40.81∼−1.27 dBm
[49]	Drone	GPS Spoofing	GNSS emulator Emulated GLONASS
[50]	Drone	GPS, RC, Video Jamming/Spoofing	Various Jamming/Spoofing
[51]	Drone	GPS Jamming	5 types of jamming
[52]	Drone	RC Jamming	Sweep jamming

**Table 4 sensors-22-02384-t004:** Studies on Non-RF interference.

Ref.	Target	Method	Signal Type/Strength
[53]	MEMS gyroscope	Acoustic	Bluetooth Speaker Distance: 10 cm SPL: 113 dB
[37]	MEMS Accelerometer	Acoustic	Speaker 2–30 kHz
[54]	Drone	Acoustic	Explosive C-4 Directed acoustic energy (pressure impulses) 3–40.6 psi
[55]	Optical flow	Projector Laser	Projector/Laser Distance: 4 feet/10 feet 138–438 lux
[30]	Lidar	Laser	- Saturating 30 mW, 905 nm (weak) 800 mW, 905 nm (strong) - Spoofing by relay Pulsed Laser Diode (PLD)

**Table 5 sensors-22-02384-t005:** Criteria for the changes in video quality by external interference signals.

Changes in Video Quality		
**Level**	**Description**	**Status**
1	No effects observed	Normal
2	Low video quality, but identifiable (Whitening, dot pattern, lagging, etc.)	Degradation
3	Unable to identify the video	Disable

**Table 6 sensors-22-02384-t006:** Experiment scenarios of low power IEMI on camera module using analog-communication scheme.

Scenario	Freq (GHz)	Distance	Power (dBm)	Detail
1	5.7∼5.8	0.3 m	23 (Max)	- Frequency sweep - Max power of signal generator - Effective frequency range analysis
2	5.74	0.3 m	−20∼11	- Power level sweep - Frequency fixed at transmission frequency - Effective signal power analysis
3	5.72∼5.75	0.3 m	10	- Frequency sweep - Power level decided in Scenario 2

**Table 7 sensors-22-02384-t007:** Experiment results of low power IEMI on camera module using analog communication scheme.

Scenario	Status	Frequency Difference ($|f_{\mathrm{channel}}-f_{\mathrm{attack}}|$, MHz)	$P_{\mathrm{out}}$ (dBm)
1	Normal	>17	23
	Degradation	15∼17	23
	Disable	≦14	23
2	Normal	-	∼−4
	Degradation	-	−5∼9
	Disable	-	10∼
3	Normal	>8	10
	Degradation	6∼8	10
	Disable	≦5	10

**Table 8 sensors-22-02384-t008:** Experiment scenarios for camera module using digital communication scheme.

Scenario	Freq	Distance (m)	Level (dBm)	Position
1	$f_{0}$: 2.43 GHz BW: 20 MHz	2.5	0∼8	Front
2	$f_{0}$: 2.33–2.53 GHz BW: 20 MHz	2.5	8	Front
3	$f_{0}$: 2.43 GHz BW: 20 MHz	2.5	0∼8	Side
4	$f_{0}$: 2.33–2.53 GHz BW: 20 MHz	2.5	8	Side

**Table 9 sensors-22-02384-t009:** Experiment results of low power IEMI on camera module using digital communication scheme.

Scenario	Status	Description	Cause
1	Degradation	- From 6 dBm - No distortion - Lagging	- No bit loss due to gradual decrease of SNR - Data rate reduction by low SNR
2	Degradation	- Center: 2.44 GHz - Serious distortion for short period - Lagging	- Bit loss due to radical decrease of SNR - Data rate reduction by low SNR
3	Normal	- No effect observed	- Small path overlap between communication and interference channel
4	Normal	- No effect observed	- Small path overlap between communication and interference channel

## Data Availability

Not applicable.

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
