# Peer review of "Review of Intentional Electromagnetic Interference on UAV Sensor Modules and Experimental Study"

_sensors, 2022, doi:10.3390/s22062384_

Round 1
Reviewer 1 Report
The reviewer thinks that the manuscript would be interesting and informative for most readers of this journal.
The following items might be corrected for better recognition:
(1) (Page 15, Line 383)
"Table reftab-camera-pix-resultsummarizes" should be
"Table 7 summarizes"
(2)(Through Clause 4.2.)
"scenario X(any number)" should be "Scenario X" (Begin by Capital S)
(3)(Titles in Chapter 4)
4.1. IMU Sensor(High power IEMI) -> IMU sensor interfered by high power IEMI source
4.2. Camera module(Low power IEMI) -> Camera module interfered by low power IEMI source
4.3. Optical flow(Non-RF) -> Optical flow sensor interfered by non-RF source
(end of comment)
Reviewer 2 Report
This review paper summarized the IEMI immunity research for UAV, it is in general well-written and of interest due to the growing interests both in terms of electromagnetic safety of UAV and anti-UAV technologies. I recommended a revision since there are issues need to be clarified.
1) the major concern was in Section 2, the review of IEMI, the author categorized IEMI as high power/lower power and non-RF, to me, although plenty of comments have been made to the three subjects, such categorization is not convincing since the concepts are obscure, given 2) a particular IEMI environment, it is hard to determine it is a high-power or lower-power IEMI.
2) From what I understood, only in-band narrowband interference can be considered as low-power IEMI? The discussions on wideband transient low-power IEMI are not sufficient. There are also literatures discussing carefully designed wideband signals with knowledge of system a prior.
3) near-field scanning is applied to find sensitive frequency; however, it is well-know electromagnetics varies dramatically in the near-field region, the conclusion might be misleading, please elaborated the effectiveness of such method.
3) Table 2, [40] is not an experiment, it is not appropriate to put it in the experiment review.
4) line 122-123, please rephrase, it is confusing.
5) Several canonical module in UAV is reviewed, for instance, camera/gps are common modules used in many electrical devices, is there any difference in terms of IEMI immunity between independent module and module mounted inside UAV, any particular observations could be interest.
Author Response
请参阅附件。

Reviewer 3 Report
Major comments:
- Fig.2, Fig.5, Fig.7, Fig.9 have poor performance and is difficult to read, please redraw them.
- Since this paper is a review paper, this article does not go far enough in presenting the existing literature. For instance, few literature has been utilized to support the classification in Table I. Please add more literatures.
Minor comment:
- The name of the laboratory is leaked in Figure 10.
Reviewer 4 Report
The reviewed article contains interesting considerations. The extensive practical analysis of issues related to the review of intentional electromagnetic interference on UAV sensor modules and experimental study, but the current version need improved, for details se below:
- line 17: change “sensors[1–6].” to “sensors [1–6].”,
- line 22: change “increasing[13].” to “increasing [13].”,
- line 24: change “neutralization[14–20].” to “neutralization [14–20].”,
- line 31: change “waves[21];” to “waves [21];”,
- line 32: change “conducted[22–25].” to “conducted [22–25].”,
- line 82: change “important[29,30].” to “important [29,30].”,
- line 86: change “antennas[31,32].” to “antennas [31,32].”,
- line 140: change “observed at 2.45 GHz .” to “observed at 2.45 GHz.”,
- Figure 1: change “the signal flow” to “the signal flow.”,
- line 296: change “known[54].” to “known [54].”,
- Figure 2: change “using ’QGroundControl’ software” to “using ’QGroundControl’ software.”,
- Figure 3: change “using log files” to “using log files.”,
- line 322: change “frequency[55],” to “frequency [55],”,
- Figure 4: change “near-field scanner” to “near-field scanner.”,
- Figure 5: change “256 MHz)” to “256 MHz).”,
- line 358: change “4.2. Camera module(Low power IEMI)” to “4.2. Camera module (Low power IEMI)”,
- line 402: change “jamming[47],” to “jamming [47],”,
- line 487 change “Conceptualization, I.-P.H.;data curation” to “Conceptualization, I.-P.H.; data curation”,
- Table 1-9: please format according to the journal's guidelines,
- References: is required Abbreviated Journal Name,
- References: include the digital object identifier (DOI) for all references where available,
- References: please format according to the journal guidelines
https://www.mdpi.com/files/word-templates/sensors-template.dot
In future publications, authors should devote more time to editing the article according to the requirements of the journal. This will then avoid quite a number of insights into editing.
The reviewed article is a valuable publication. It can serve readers as a set of knowledge that can be used as a basis for further innovative and implementation studies.
Round 2
Reviewer 2 Report
The authors have clarified my concerns, just a final comment: please add suitable label and unit to axis of figure 13.